# Iso-acoustic focusing of cells for size-insensitive acousto-mechanical phenotyping

Per Augustsson[1,2], Jonas T. Karlsen[3], Hao-Wei Su[1], Henrik Bruus[3] & Joel Voldman[1]

Mechanical phenotyping of single cells is an emerging tool for cell classification, enabling assessment of effective parameters relating to cells' interior molecular content and structure. Here, we present iso-acoustic focusing, an equilibrium method to analyze the effective acoustic impedance of single cells in continuous flow. While flowing through a microchannel, cells migrate sideways, influenced by an acoustic field, into streams of increasing acoustic impedance, until reaching their cell-type specific point of zero acoustic contrast. We establish an experimental procedure and provide theoretical justifications and models for iso-acoustic focusing. We describe a method for providing a suitable acoustic contrast gradient in a cell-friendly medium, and use acoustic forces to maintain that gradient in the presence of destabilizing forces. Applying this method we demonstrate iso-acoustic focusing of cell lines and leukocytes, showing that acoustic properties provide phenotypic information independent of size.

[1] Department of Electrical Engineering and Computer Science, Massachusetts Institute of Technology, 77 Massachusetts Avenue, Cambridge, MA 02139, USA. [2] Department of Biomedical Engineering, Lund University, Ole Römers väg 3, 22363, Lund, Sweden. [3] Department of Physics, Technical University of Denmark, DTU Physics Bldg 309, DK-2800 Kongens Lyngby, Denmark. Correspondence and requests for materials should be addressed to P.A. (email: per.augustsson@bme.lth.se).

Microfluidic methods to position, separate, and analyze cells hold promise to shorten the time from sample to answer in cell-based assays in health care and basic biology[1,2]. In particular, phenotyping of single cells based on their cell-intrinsic biophysical properties is an emerging tool for classification of cells that relates to differences or changes in their molecular and architectural structure[3]. To date the palette of cell properties made available for analysis in microfluidic systems include size[4], density[5], deformability[1,6,7], electrical[8,9], and optical[10] properties.

Acoustic characterization of tissue has tremendous value in medicine through various forms of medical ultrasound imaging applications such as sound scattering, attenuation and elasticity[11,12]. It is therefore reasonable to assume that acoustic properties of individual cells, which include mass density and adiabatic compressibility, are meaningfully related to their biological phenotype. Mapping of sound velocity and attenuation at sub-cellular resolution has been demonstrated for adherent cells using acoustic microscopy[13–16] indicating that the acoustic properties are related to the interior content and structure of a cell. While the population average of the acoustic properties of cells in suspension can be deduced from measurements using standard density and sound velocity meters[13–17] the acoustic properties of single suspension cells are largely unknown.

Acoustophoresis[18], relying on acoustic radiation forces, has emerged as a gentle[19,20] and robust method for concentrating[21,22], trapping[23,24], washing[25], aligning[26] and separating cells[27–29]. However, the strong size dependency in acoustophoresis has hampered the development of cell analyzers based on measuring the underlying properties of cell density and compressibility, a shortcoming shared by all volumetric force-based methods. Although examples exist of canceling the first-order size dependency via balancing against gravity[30], loading cells with immuno-affinity microbeads[31,32], or by estimating and adjusting for cell size[33], no acoustophoretic method exist today that allows size-independent cell analysis of individual cells in suspension.

Here we introduce a size-insensitive method, iso-acoustic focusing (IAF), that can analyze cells based on the previously uncharted parameter of effective acoustic impedance. This equilibrium method can be viewed as a microfluidic analog to density gradient centrifugation or iso-electric focusing. Cells flowing through a microchannel migrate sideways, influenced by an acoustic field, into flow streams of ever increasing acoustic impedance. Finally, the individual cells reach their iso-acoustic point (IAP), at which the acoustic contrast between the cell and the surrounding liquid becomes zero, and the sideways displacement ceases. Cell-specific differences in effective acoustic impedance translate to a spatial dispersion of the cell population transverse to the flow, enabling continuous label-free analysis of individual cells. To develop this method, we have first identified a suitable molecule (iodixanol) to alter the acoustic properties of the cell-culture medium such that cells can have positive, zero or negative acoustic contrast depending on the molecular concentration. We utilize here a recent finding that acoustic impedance gradients are self-stabilized in resonant acoustic fields, which counteracts any gravitational relocation of the laminated liquids due to density differences[34].

We lay out a simple theoretical model of the acoustofluidic system describing the shaping of a smooth impedance gradient through interactions of diffusion, gravity and acoustic radiation, and compute the trajectories of cell migration in the resulting acoustofluidic field. We then characterize the IAF system using cell lines and apply it to measuring the effective acoustic impedance of white blood cells.

## Results

**IAF Principle.** In IAF, cells are deflected sideways by a half-wavelength resonant acoustic pressure field $p(y, t)$,[35–37] oriented orthogonal to the flow (Methods—Measuring the acoustic field, Supplementary Fig. 1), in a laminar flow microchannel (Fig. 1a). Cells suspended in a standard cell-culture medium are injected near both side walls of the channel and cell-free liquid of higher acoustic impedance $Z_{med}$ is injected in a central inlet to occupy the central part of the flow. The flow velocity of the channel is tailored such that a smooth profile $Z_{med}(y)$ with an associated acoustic impedance gradient forms by way of molecular diffusion. The transverse acoustic radiation force $F_{rad}$ on a given cell stems from momentum transfer to the cell from the sound wave due to scattering. Because this scattering is governed by differences in mass density $\rho$ and adiabatic compressibility $\kappa$ between the cell and the surrounding medium, there exists a medium condition for which the acoustic contrast $\Phi$ and force $F_{rad}$ are zero, and thus the acoustically induced sideways velocity $u_{rad}$ vanishes. This condition we refer to as the iso-acoustic point (IAP), (Fig. 1b). To a good approximation the IAP is the location at which $Z_{med}$ equals the effective acoustic impedance $Z_{cell}$ of the cell (Supplementary Note 1). Since the sound wavelength is in IAF much longer than the size of a single cell the effective acoustic impedance can be interpreted as a measure of the integral of the interior variations in acoustic properties that has been previously mapped using acoustic microscopy[14–16].

A cell initially near a wall at $y = 0$ will migrate toward the channel center due to a positive acoustic contrast. Upon traversing up the concentration or impedance gradient of the medium, the acoustic contrast eventually becomes zero at the IAP, preventing the cell from moving any further. The configuration is stable in the sense that if the cell starts out in the channel center, it will instead move out towards the walls, down the impedance gradient, until reaching the same IAP. When reaching the end of the microfluidic channel, the sideways position of individual cells can be recorded and then translated to an effective cell acoustic impedance, since at the IAP $Z_{cell} = Z_{med}$.

**Tuning the acoustic contrast between medium and cells.** Critical to IAF is the ability to prepare separation media of acoustic impedance higher and lower than that of the cells, thus

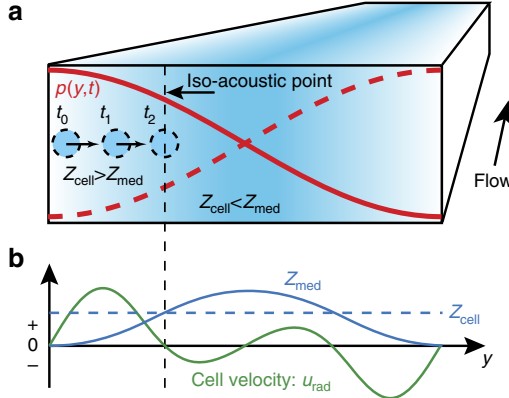

**Figure 1 | The IAF principle. (a)** Cells (circles) flowing in a microchannel are deflected sideways towards the node of an acoustic resonant pressure field $p$ (red curves) in a medium of position-dependent acoustic impedance $Z_{med}$ (color plot from low (white) to high (blue)). **(b)** Conceptual plot showing that when the acoustic impedance $Z_{cell}$ (dashed blue line) of a given cell matches $Z_{med}$ (full blue line) at the IAP, its transverse velocity $u_{rad}$ (green line) becomes zero so that its position along $y$ reflects its individual effective acoustic impedance.

enabling a transition from positive to negative acoustic contrast. The vast majority of cells have positive acoustic contrast in standard cell-culture media (a notable exception being adipocytes[38]). To increase the acoustic impedance of the medium, OptiPrep, a standard isotonic gradient centrifugation medium that contains iodixanol, was added. Iodixanol was chosen by virtue of its high acoustic impedance to viscosity ratio (Supplementary Fig. 2c and 2e), allowing substantial changes in acoustic impedance with only modest changes in viscosity. The acoustic impedance of iodixanol solutions increases monotonically with concentration when using isotonic serum-supplemented saline solutions as a diluent, such that the cells are expected to have positive acoustic contrast when suspended in low-concentration iodixanol and to have negative contrast for high-concentration iodixanol.

To demonstrate that the acoustic contrast of cells can be modulated by altering the properties of the medium we suspended murine BA-F3 pro B cells in a range of iodixanol concentrations, and observed their trajectories when exposed to a resonant sound field. Cells were injected in the acoustic microfluidic channel, the flow was stopped, and the sound was turned on while recording the trajectories of the cells (Supplementary Movies 1-3). The trajectories were then analyzed and classified (Methods—Stop flow trajectory classification, Supplementary Fig. 3) as having positive contrast if they moved to the central node and negative if they moved to either side-wall. 96% of cells suspended in 10% iodixanol had positive contrast, whereas most cells (83%) in 25% iodixanol had negative contrast (Fig. 2). For intermediate concentrations, many cells display oscillatory motion patterns (zero contrast), which indicates that the cells are predominantly influenced by acoustic streaming rather than acoustic radiation. Acoustic streaming is a phenomenon associated with acoustic fields in fluids that here induces a slow rotation of the bulk liquid[39–41]. Cells having zero acoustic contrast will experience a drag force from the rotating liquid exceeding that of the acoustic radiation force acting directly on the cell while cells of positive and negative contrast will be radiation dominated.

These results show that one can create both positive and negative acoustic contrast by tuning medium properties with iodixanol. Additionally, by assuming $Z_{cell} = Z_{med}$ at zero acoustic contrast, we can further conclude that these BA-F3 cells have acoustic impedances between $1.6\,\mathrm{MPa\,s\,m^{-1}}$ and $1.7\,\mathrm{MPa\,s\,m^{-1}}$.

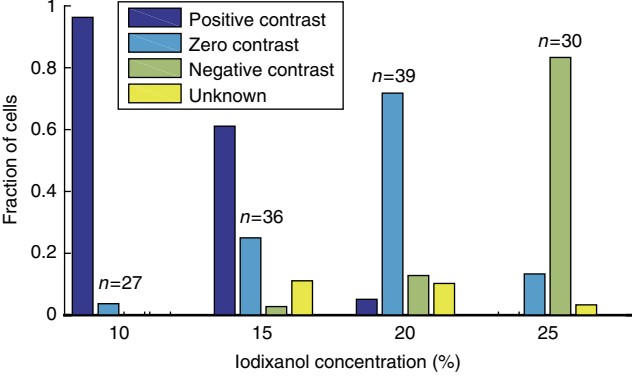

**Figure 2 | Controlling the acoustic contrast of BA-F3 cells by altering the acoustic properties of the suspending medium.** Fraction of cells exhibiting *Positive, Negative* or *Zero contrast* based on multi frame trajectory analysis. *Unknown* refers to cell trajectories that did not match any of the other categories. *n* refers to number of cell tracks, with a minimum five onsets of sound.

**Creating an acoustic impedance gradient**. To separate cells of different acoustic properties to distinct, stable locations in the acoustic field, the acoustic impedance of the liquid must form a spatial gradient that is monotonically increasing toward the channel center. We achieve this by laminating high acoustic impedance media containing iodixanol side by side with standard cell media of lower acoustic impedance. The steep acoustic impedance gradient at the inlet becomes flattened by diffusion of the iodixanol molecules during the flow through the channel. Injecting a fluorescent dextran (MW 3000 Da) tracer of similar molecular diffusion constant ($D_{dex} \approx 2.2 \times 10^{-10}\,\mathrm{m^2\,s^{-1}}$)[42] as iodixanol ($D_{ix} \approx 2.5 \times 10^{-10}\,\mathrm{m^2\,s^{-1}}$)[43] allows indirect visualization of the iodixanol concentration profile.

First consider a homogeneous solution (20% iodixanol) with no density gradient. A dilute fluorescent tracer gradient is created by injecting different concentrations of tracer in the central versus side inlets. It broadens due to molecular diffusion and flattens in more than 68 s (Supplementary Fig. 4c). Repeating this experiment, but increasing the concentration of iodixanol to 30% in the center layer while reducing the concentration to 10% in the side inlets induces $\sim 30$-fold faster flattening of the fluorescent tracer profile ($\sim 2\,\mathrm{s}$, Supplementary Fig. 4b). This is caused by gravitational collapse of the high-density central liquid layer, which ends up in a stable configuration along the channel floor as confirmed by confocal imaging (Fig. 3a). Finally, repeating the latter experiment, but with ultrasound on, acoustic radiation forces act on the central liquid layer and stabilize it against gravitational collapse (Fig. 3b), an effect previously observed by confocal microscopy[34]. The condition for stability is that the acoustic energy density is larger than the hydrostatic pressure (Supplementary Note 2). The acoustically stabilized concentration profile still broadens over more than 34 s due to diffusion (Supplementary Fig. 4a) similar to the homogeneous-density system without ultrasound (Supplementary Fig. 4c).

As was pointed out in the previous section, acoustic streaming is observed for cells of zero acoustic contrast in the case of homogeneous medium. In the inhomogeneous system we observed an iodixanol concentration profile that evolved over more than 34 s (Supplementary Fig. 4a) which indicates that acoustic streaming is not present in the bulk to the extent that it affects the acoustic impedance gradient. We explain this absence of streaming in the bulk by a scaling argument presented in Supplementary Note 2. In essence, the acoustic force density stabilizing the gradient can be shown to be orders of magnitude larger than the shear-force density associated with the boundary-driven acoustic streaming. We can make the stabilized acoustic impedance gradient steeper or shallower by tuning the overall volume flow rate in the channel and thereby controlling the time available for diffusion of the iodixanol and the fluorescent tracer before the downstream imaging region is reached (Fig. 3c). This allows tuning of the range and resolution of the system so as to maximize the spatial spread of cells' IAP for a given input sample. Further, the position of a cell's IAP can be controlled by varying the relative flow rates of the side and central inlet streams (Fig. 3d).

In summary, we can create smooth acoustic impedance gradients and tailor their range according to the anticipated IAP of different cells. Remarkably, the same acoustic field that stabilizes these gradients also drives the cells towards their IAP.

**Measuring the effective acoustic impedance of cells**. To measure the acoustic impedance of individual cells in continuous flow, fluorescently labeled cells were resuspended in low-impedance medium (10% iodixanol) and injected into the side inlets while high-impedance media (iodixanol 36 %) containing dextran

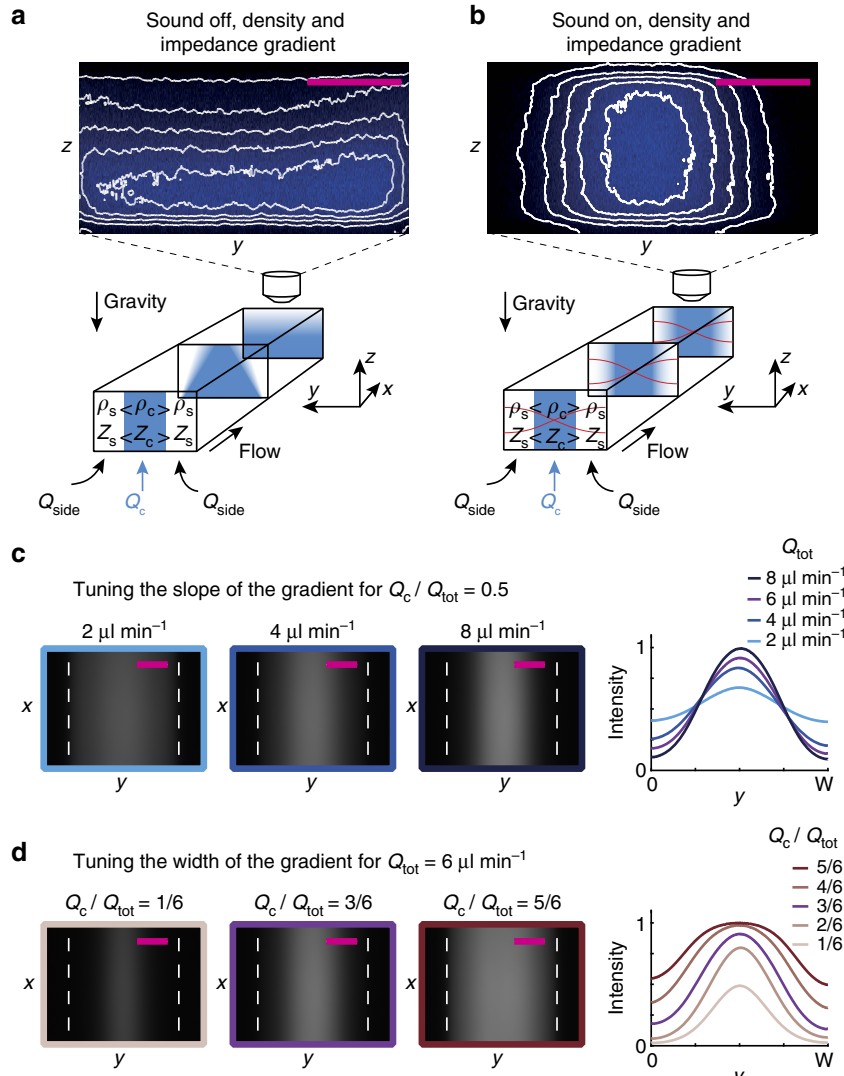

**Figure 3 | Tailoring smooth acoustic impedance gradients. (a** and **b**) Confocal cross sectional $y$-$z$ scans recorded 20 mm downstream from the inlet, of the fluorescent dextran tracer gradient (blue color plot) in the case of (**a**) an iodixanol gradient and no sound, and (**b**) an iodixanol gradient and sound (red lines), along with schematic interpretations of the data. (**c**) Top views of the fluorescent dextran tracer gradient imaged with epifluorescence microscopy at different volume flow rates $Q_{tot}$. The graph shows the corresponding normalized fluorescence intensity profiles versus $y$ averaged along the flow direction $x$. (**d**) Top view epifluorescence images when varying the relative flow rate in the central $Q_c$ and side inlets while maintaining a constant overall flow rate $Q_{tot}$ and the corresponding normalized fluorescence intensity profiles. Scale bars (magenta) are 100 µm.

tracer was injected through the central inlet (Fig. 4a). The resonant acoustic field pushes cells towards higher acoustic impedance in the self-forming concentration profile until they reach their respective IAP. At the end of the channel, by repeatedly imaging the dextran tracer gradient and then the cells, we could infer the cells' IAP from the local dextran fluorescence intensity—and thus the concomitant iodixanol concentration and acoustic impedance $Z_{med}$ (Fig. 4b-e). The method is described in more detail in Methods—Method to measure cell IAP.

We introduced BA-F3 cells into the device and measured their IAP and inferred their acoustic impedances $Z_{cell}$ (Fig. 4e). We found that the measured acoustic impedance $Z_{cell} = 1.68$ MPa·s m$^{-1}$ was stable over time with no significant drift (Fig. 4f, slope = 3.04 Pa m$^{-1}$, correlation coefficient r = 0.039, CI$_{95}$: − 0.013, 0.090, $P = 0.14$ computed with t-statistic). Comparing the populations of cells measured on the right and left hand side of the channel reveals a small difference of the means of 0.005 MPa s m$^{-1}$ (CI$_{95}$: 0.0040, 0.0069) which is less

than half the sample standard deviation 0.014 MPa s m$^{-1}$ and thus insignificant.

To verify that the cells have reached their IAP upon entering the imaging region, the flow was kept constant while varying the piezoceramic actuator voltage amplitude $U$ and thereby the acoustic energy density $E_{ac}$ and acoustic pressure amplitude $p_a$ ($E_{ac} \propto p_a^2 \propto U^2$, Methods—Measuring the acoustic field, Supplementary Fig. 5). Measurements of the apparent cell effective acoustic impedance distributions showed that the distributions narrowed and approached the same median value (Fig. 4g) at acoustic energy densities above $E_{ac} = 11$ J m$^{-3}$. This behavior is consistent with an equilibrium separation method. In addition, reducing the overall flow rate from 8 µl min$^{-1}$ to 4 µl min$^{-1}$ while maintaining the same actuation settings does not alter the measured distributions markedly (Fig. 4h), which shows that the method is insensitive to flow variations, again consistent with an equilibrium method. The inferred acoustic impedances of BA-F3 cells range from 1.66 MPa s m$^{-1}$ to

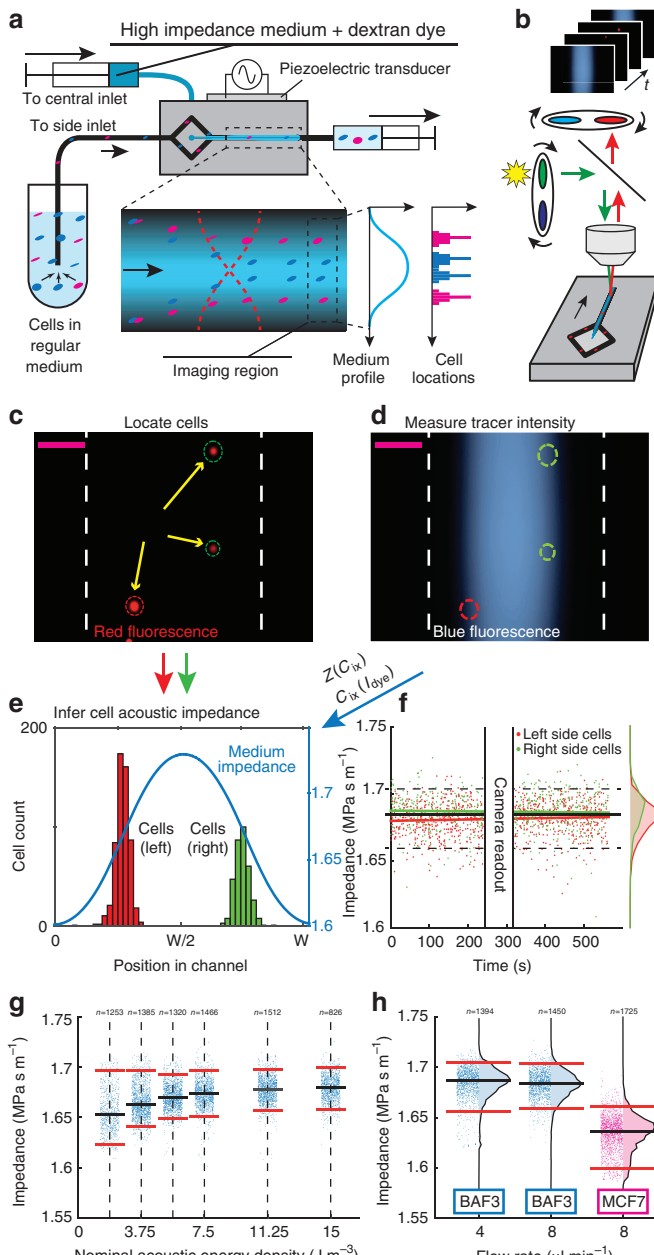

**Figure 4 | IAF cell measurements.** (**a**) Schematic of the system. (**b-e**) The acoustic impedance gradient is inferred via (**b**) sequential imaging at the end of the channel of (**c**) cells and (**d**) the fluorescent dextran tracer gradient. Scale bars (magenta) are 100 μm. (**e**) Standard solutions of iodixanol and dextran dye were analyzed to convert from fluorescence intensity to acoustic impedance. (**f**) Scatter plot during 500 s of the inferred acoustic impedance $Z_{cell}$ of BA-F3 cells ($n = 1,450$) passing the imaging region in both sides of the device. The red and green lines show the linear least square fits. The solid black horizontal line shows the median of all cells and the dashed lines show the 5th and 95th percentiles. (**g**) Scatter plots of $Z_{cell}$ for BA-F3 ($n = 7,762$) for six values of the acoustic energy density. (**h**) Scatter plots of $Z_{cell}$ for BA-F3 cells at flow rates 4 μl min$^{-1}$ ($n = 1,394$) and 8 μl min$^{-1}$ ($n = 1,450$) and for MCF7 cells at 8 μl min$^{-1}$ ($n = 1,725$). Black horizontal lines show median and red lines show the 5th and 95th percentiles.

1.70 MPa s m$^{-1}$ (5th to 95th percentile), which is within the range found by the stop-flow experiments in Fig. 2.

To see if the IAP is cell-type specific, we measured human breast cancer cell line MCF7 cells (Fig. 4h) and found that the

measured acoustic impedance was lower than for the BA-F3 cells with only a minimal overlap of the two populations such that only 6.07% of the MCF7 cells were found above the 5th percentile of the BA-F3 cells. Because the diameter of MCF7 cells is ~1.5 times larger than that of BA-F3, and, in this system, lower acoustic impedance implies that cells are detected closer to the channel walls; this result also underpins the size insensitivity of IAF in being an equilibrium method. In traditional non-equilibrium acoustophoresis cell separation, where larger cells move faster than smaller cells, the MCF7 cells would end up closer to the channel center for a given time of exposure to sound.

To validate that cells indeed reach and stay in their IAP given the transit time through the IAF channel, and to illustrate the method in more detail, we implemented a simple numerical model of cells migrating in acoustic fields in diffusing iodixanol gradients (Supplementary Note 3, Supplementary Fig. 6). We modeled the trajectories of a BA-F3 cell and a MCF7 cell based on the measured effective acoustic impedances from the experiments, the literature values for density, and their sizes from Coulter counter measurements. We also performed a time-scale analysis of the process to give analytical estimates for deciding experimental design parameters (Supplementary Note 4). In brief, both cells reach their respective IAP within the first 5.7 s and stay there for up to 51 s after entering the channel, which corresponds to flow rates ranging from 1.3 μl min$^{-1}$ to 12 μl min$^{-1}$. The diffusing gradient causes the position of the BA-F3 cell's IAP to approach the channel center over time while the MCF7 cell eventually ends up near a wall. The diffusing gradient narrows the measurement range leading to finer resolution in the IAP measurement as the profile evolves. A cell reaching the channel center or a wall indicates that the cell has an effective acoustic impedance that is higher or lower, respectively, than the upper or lower bounds on the measurement range.

**Blood cell characterization.** To understand to what extent acoustic impedance can discriminate cells from mixed populations, we analyzed primary human monocytes, lymphocytes and neutrophils purified from whole blood by negative selection (Fig. 5a). Lymphocytes (red) and monocytes (blue) have partly overlapping acoustic impedance distributions, whereas neutrophils (gray) have substantially higher acoustic impedance. The results suggest that neutrophils can be distinguished from lymphocytes and monocytes purely based on their location in an acoustic impedance gradient. Since lymphocytes and monocytes have substantially different sizes (~7.5 μm and ~9.0 μm, respectively, Fig. 5e) but similar acoustic impedance (~1.69 MPa s m$^{-1}$, Fig. 5c), whereas neutrophils and monocytes have similar sizes (~9.0 μm, Fig. 5e) but differing acoustic impedances (~1.73 MPa s m$^{-1}$ and ~1.69 MPa s m$^{-1}$, respectively, Fig. 5c), these results further illustrate the size-independence of the IAF method.

To highlight how acoustic properties combined with optical measurements can form a two-parameter classification analogous to flow cytometry, without using cell type-specific labels, we point to the scatter plot and the associated distributions in Fig. 5b-d of the effective acoustic impedance vs the total cell fluorescence intensity. Even though the total intensity is not a true measure of cell volume, the scatter plot allows us to distinguish monocytes from lymphocytes based on an optical measurement, while the assessment of the effective acoustic impedance enables identification of neutrophils.

Furthermore, the sensitivity of IAF is high enough to detect alterations in mechanical properties of cells. Analysis of RBC-lysed human blood reveals a distinct peak (purple) corresponding to neutrophils, which are normally the most

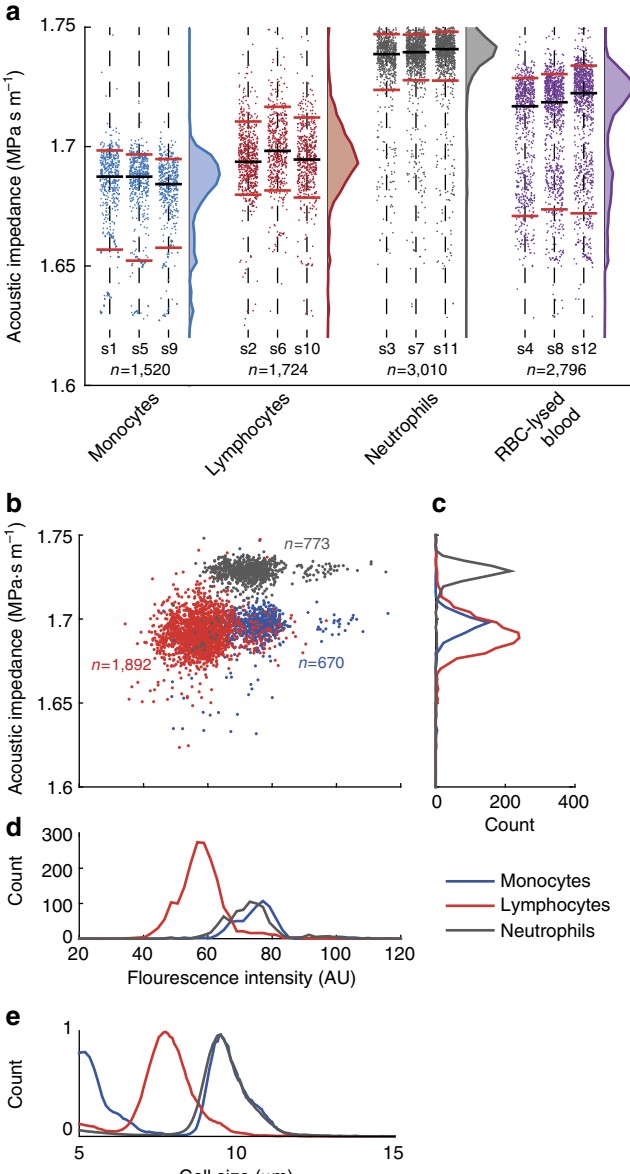

**Figure 5 | The effective acoustic impedance of white blood cells as measured by their IAP in an iodixanol gradient.** (**a**) Scatter plots containing 9050 data points of subsequent measurements of pre-enriched monocytes (blue), lymphocytes (red) and neutrophils (gray) isolated by negative depletion, and of white blood cells isolated by RBC-lysis of whole blood from a single donor. Labels s1 to s12 indicate the order of the sample analysis. Distributions show the sum of all measured cells for the three repeats. Black horizontal lines show the median and red lines show the 5th and 95th percentiles. (**b**) Scatter plots of the measured effective acoustic impedance vs fluorescence intensity of pre-enriched monocytes, lymphocytes and neutrophils from a second donor, and the distributions of (**c**) the effective acoustic impedance and (**d**) fluorescence intensity. (**e**) Size distributions of monocytes, lymphocytes and neutrophils as measured independently based on electrical resistance sizing (Coulter-counter).

abundant of the white blood cells (~60%), Fig. 5a. We find that this peak is shifted towards lower effective acoustic impedance compared to the peak (gray) of the neutrophils purified directly from whole blood. It is reasonable to assume that this shift is due to altered physical properties of the white blood cells when exposed to the RBC-lysis and subsequent centrifugation steps.

## Discussion

We have presented and provided experimental and theoretical validations of the iso-acoustic focusing (IAF) equilibrium method for measuring the effective acoustic impedance of individual cells in continuous flow. IAF was applied to measure the effective acoustic impedance of monocytes, lymphocytes, neutrophils, BA-F3 and MCF7 cells. The IAPs of thousands of individual suspension cells were measured in just a few minutes and we could load and run up to 12 sequential samples within 3 hours, which make this a suitable method for high throughput intrinsic phenotyping of cells.

While cell size is an important parameter for categorizing and separating cells, it displays large variations across cell types. We have demonstrated in several experiments that IAF is size insensitive. For instance, the populations of monocytes and neutrophils differ markedly in acoustic impedance (Fig. 5a) despite almost completely overlapping size distributions (Fig. 5e).

We find it relevant to further investigate cell IAP as a potential biomarker since it was found to be cell-type specific with measured effective acoustic impedances ranging from 1.55 MPa s m$^{-1}$ to 1.75 MPa s m$^{-1}$ (Fig. 4h and Fig. 5a). Although organs and tissue typically consist of multiple cell types which can in turn have fundamentally different internal structure than the cells under investigation here we find it interesting to note the close correspondence to literature values for measured acoustic impedances of various tissues (fat 1.38 MPa s m$^{-1}$, brain 1.60 MPa s m$^{-1}$, heart 1.45 MPa s m$^{-1}$, kidney 1.65 MPa s m$^{-1}$, blood 1.66 MPa s m$^{-1}$, liver 1.69 MPa s m$^{-1}$, skin 1.99 MPa s m$^{-1}$)[44] with fat, heart and skin being outside our established range. Further, the measurements on white blood cells (Fig. 5a) and the MCF7 cancer cells (Fig. 4h) indicate that these cancer cell line cells can be separated from blood cells with high purity based on their lower IAP. While no clinical samples has been analyzed in this study, this suggests that IAF can potentially be employed to isolate circulating tumor cells from cancer patient blood.

The acoustic properties of suspended MCF7 cells have previously been measured, by tracking individual cells in an acoustic field in stopped-flow homogeneous media[33], to have an effective acoustic impedance of 1.59 MPa s m$^{-1}$, and for adherent cells using acoustic microscopy[15], to 1.56 MPa s m$^{-1}$, both of which are in the lower range of the IAP$_{MCF7}$ ($1\sigma$ and $2\sigma$ below the mean, respectively) shown in Fig. 4h. Previously acoustic levitation against gravity has enabled measurement of the acoustic impedance of RBCs (1.81 MPa s m$^{-1}$)[30], just above the IAP of neutrophils (Fig. 5a) which is not surprising given the slightly higher density of RBCs compared to neutrophils.

Cell deformability has been extensively studied for suspension cells as well as adherent cells and is informative about interior structure[3]. In contrast, IAF relates to the whole–cell, or effective, compressibility and density, and we believe it renders complementary acousto-mechanical information. As we have seen the effective impedance of neutrophils was dependent on exposure to lysis buffer and we can therefore speculate that IAF can be useful for assessing cell state and disease progress. Microfluidic single cell density measurements have been previously demonstrated to enable sensitive monitoring of yeast growth cycle[45] and altered cell state resulting from environmental perturbations[5]. We envision that IAF can be informative in a similar way but benefit from higher throughput and flexibility in terms of sample preparation and integration with complementary microfluidic operations. IAF could for instance be combined with label-free or label-based optical analysis enabling an iso-acoustic

tunable band-pass gate within which we interrogate cells of interest.

The results show that, in IAF, stratified liquids and suspended cells arrange themselves in a way that can be predicted from their effective acoustic impedances. This rearrangement and process is analogous to another well-established equilibrium method, density gradient centrifugation, but IAF has the benefit of being straightforwardly implemented in microfluidic systems. The precise spatiotemporal control offered by microfluidics[46,47] leads us to envision that the IAF method can be developed into a tunable cell profiling method as well as a new tool for size-insensitive label-free cell separation.

## Methods

**Device fabrication.** The IAF chip was fabricated by UV-lithography and aniso-tropic KOH wet etching a channel structure in <100> silicon rendering a rectangular cross-section acoustic resonator channel (25 mm long, 375 μm wide, 150 μm deep), Supplementary Fig. 7a. The channel has a trifurcation inlet where cell-free media can be introduced through a central branch and cell suspension can be introduced via a common side inlet port that bifurcates to either side of the central branch. Fluid access holes were etched from the back side of the chip. Borosilica glass was anodically bonded to the silicon to seal the channel structure[48].

**Acoustic actuation.** A piezo-ceramic actuator was bonded to the back of the chip (Supplementary Fig. 7b) and connected to a function generator. In the blood cell experiments, to further increase the ultrasound intensity, a radio frequency amplifier (TVA-R5-13, Mini-Circuits, Brooklyn NY, USA) was connected in series with the output of the function generator. The resulting transducer amplitude peak to peak was measured with an oscilloscope to be 6.0 V (leading to an acoustic field amplitude in the channel of $E_{ac} = 43$ J m$^{-3}$, $p_a = 0.66$ MPa) in the blood cell measurements and 3.5 V (leading to an acoustic field amplitude in the channel of $E_{ac} = 15$ J m$^{-3}$, $p_a = 0.39$ MPa) in the cell line experiments. The function generator was set to make repeated linear sweeps in frequency from 1.900 MHz to 2.100 MHz over periods of 1 ms. This leads to stable operation and a more even acoustic field along the length of the device[49].

**Flow setup.** Liquid to the central inlet was pumped from a glass syringe mounted in a syringe pump and cell suspension was pushed into the side inlet port from the bottom of a pressurized test tube (10 psi). The outlet flow from the acoustic channel was controlled by a syringe pump (Fig. 4a). In all cell measurements a 1:1 flow ratio was maintained between the central and side inlet flow streams.

**Medium to alter acoustic impedance.** A stock solution of 20% iodixanol was prepared from 1 ml OptiPrep (contains 60% iodixanol) by addition of $2 \times 1$ ml of DPBS supplemented with 2 % fetal bovine serum (FBS) by reversed pipetting.

A stock solution containing 36 % iodixanol was prepared in the same manner by adding $3 \times 1$ ml OptiPrep and $2 \times 1$ ml DPBS with 2 % FBS, using reversed pipetting. 3 μg ml$^{-1}$ fluorescent tracer molecules, Dextran Cascade Blue 3000 Da, was added.

**Cell culture.** The cells were cultured using standard protocols at 37 °C in a humidified atmosphere with 5% CO$_2$. All media were supplemented with 10% fetal bovine serum (FBS) and Penicillin 100 U ml$^{-1}$, Streptomycin 100 μg ml$^{-1}$.

BA-F3 murine pro B cells (Deutsche Sammlung von Mikroorganismen und Zellkulturen) were grown in a cell culture flask in RPMI 1640 media supplemented with Interleukin-3 (1 ng ml$^{-1}$) and L-glutamine.

MCF7 human breast cancer cells (American Type Culture Collection) were grown in a cell culture dish in low glucose DMEM (Life Technologies). Cells were detached by trypsin/EDTA treatment and subsequent resuspension in DPBS supplemented with 2 mM ethylenediaminetetraacetic acid (EDTA).

**Blood cell sample preparation.** Blood was obtained from healthy volunteers with informed consent and was kept at room temperature in test tubes containing (EDTA) as anticoagulant. Sample preparation was initiated within 2 hours from blood draw.

Monocytes from 2 ml whole blood were isolated by immunomagnetic negative selection using the Direct Human Monocyte Isolation Kit (EasySep, Stemcell technologies) according to the manufacturer instructions. Similarly, lymphocytes and neutrophils were isolated from aliquots of 1 ml whole blood using the Direct Human Total Lymphocyte Isolation Kit and the Direct Human Neutrophil Isolation Kit, respectively.

Leukocytes were prepared from 500 μl whole blood by standard isotonic lysing of RBCs based on ammonium chloride according to the manufacturer instructions. To remove platelets, cells were resuspended two times in DPBS supplemented with 2 mM containing EDTA.

Size distributions of cells from each fraction was measured based on electrical impedance measurements (Multisizer II, Beckman Coulter), Fig. 5e. Cells from each fraction were stained for CD45, CD66B, CD14 and CD3 and analyzed in a flow cytometer (Accuri C6, BD Biosciences) to assert purities: monocytes with 89.9 % purity, granulocytes with 94.1 % purity, and lymphocytes with 82.8 % purity (Supplementary Fig. 8).

Cells were resuspended in 1 ml DPBS supplemented with 2 mM containing EDTA and incubated for 20 min with Calecein AM orange red (2 μg ml$^{-1}$). Cells were then resuspended two times in 1 ml DPBS supplemented with 2 mM EDTA and 2 % Fetal Bovine Serum and kept on ice until the introduction into the acoustic chip.

**Measuring the acoustic field.** To measure the shape of the local acoustic field along a 1 mm segment of the IAF channel, fluorescent polystyrene microbeads (radius 2.23 μm) were investigated using micro particle image velocimetry (PIV). Particles were suspended in 10 % iodixanol (homogeneous) and injected in the IAF channel and the flow was stopped before turning on the acoustic field for a transducer voltage of 3.5 V. Image sequences were acquired at 4 Hz capturing the motion of the microbeads in the acoustic field and commercial PIV software (Dynamic studio, Dantec Dynamics, Denmark) was used to extract velocity information from the images (Supplementary Fig. 1a) using the method of adaptive correlation. The $y$-components (transverse to the channel) of the velocity vectors from 5 consecutive frames for three repeated acoustic actuation experiments are plotted in Supplementary Fig. 1b. Data was fitted (black dashed line) to the expression $u_{rad} = u_0 + \partial_t y$ following Eq. (12) in Supplementary Note 3, using $u_0$, $E_{ac}$ and $k_y$ as fitting parameters for a spherical polystyrene particle of radius 2.23 μm, density 1050 kg m$^{-3}$ and compressibility $1.65 \times 10^{-10}$ Pa$^{-1}$. Medium properties for 10% iodixanol (Supplementary Fig. 2) were used to calculate that the acoustic contrast factor $\Phi = 0.199$. The fit resulted in $u_0 = -3.557 \times 10^{-7}$ m s$^{-1}$ (CI$_{95}$: $-8.014 \times 10^{-7}$, $9.004 \times 10^{-8}$), $E_{ac} = 16.27$ J m$^{-3}$ (CI$_{95}$: 16.16, 16.39), and $k_y = 8315$ m$^{-1}$ (CI$_{95}$: 8293, 8336). The fitted acoustic wave vector $k_y$ corresponds to a sound wavelength $\lambda = 2\pi/k_y = 756$ μm which is indicative of a half wavelength resonance in the 375 μm wide IAF channel. The fitted local acoustic energy density at 3.5 V actuation corresponds to a local pressure amplitude within the field of view of $p_a = 0.39$ MPa.

To estimate the acoustic pressure amplitude and the average acoustic energy density along the whole length of the acoustic resonator channel, as a function of the transducer voltage amplitude, microbeads suspended in 10% iodixanol were injected through a single inlet at constant flow rate. The final position of the microbeads as they arrive at the end of the channel depends on the acoustic energy density, the size and acoustic properties of the particle, and the viscosity, the flow profile and the acoustic properties of the liquid[39]. The trajectories of polystyrene microparticles in acoustic fields are well characterized in earlier work[35,36], and by comparing simulation to experiment we could estimate the average acoustic energy density. The beads were imaged 20 mm downstream from the inlet with the microscope focus set at the mid height of the channel. From a sequence of such images, we estimated how far into the channel the outermost beads had moved when entering the interrogation region, Supplementary Fig. 5a-e. Then we ran the script iteratively and updated the acoustic energies for each voltage setting until the simulations for each case matched the experimental observations, Supplementary Fig. 5f. Thereafter we made a fit for $E_{ac} = \frac{1}{4}\kappa_m p_a^2 = kU^2$, Supplementary Fig. 5g, and concluded that $k = 1.2$ J m$^{-3}$ V$^{-2}$ and that $E_{ac}^{3.5V} = 15$ J m$^{-3}$ for a transducer voltage amplitude of 3.5 V.

**Stop-flow cell trajectory classification.** Images of 30 consecutive frames were analyzed, tracking the cell positions using the video analysis tool Tracker (http://physlets.org/tracker/). Supplementary Figure 3a-c show the cell images from all the frames in the repeated experiments projected onto a single image for three different concentrations of iodixanol. Only tracks of 20 frames or longer were analyzed.

First a cell was classified to have *zero* acoustic contrast if it displayed a track that is indicative of following the acoustic streaming of the bulk liquid rather than migrating to either the channel center or towards the side walls. This means the velocity changes sign or the cell flow past the channel center but do not stop there. Specifically, the starting position is away from the channel center and the channel walls and one of the following is true: The velocity changes sign and the ratio of minimal to maximal velocity is greater than 20%, or the cell crosses over the central region but does not stop within the central region.

Second, a cell of *positive* acoustic contrast was any cell that had not been classified as having *zero* acoustic contrast, and that starts out away from the channel center but ends up within the central region.

Third, a cell of *negative* acoustic contrast was any cell that had not been classified as having *zero* acoustic contrast, and that starts out away from the channel walls but ends up near either wall.

Cells that were not classified as any of the above were classified as *unknown*.

**Method to measure cell IAP.** Prior to each IAP measurement 100 μl of cell suspension was mixed with 100 μl of 20% iodixanol stock solution to a final concentration of 10% iodixanol in a test tube that was connected to the side inlet of

the acoustic microchannel. High acoustic impedance medium containing 36% iodixanol and fluorescent dextran tracer was loaded in a syringe connected to the central inlet, Fig. 4a.

The channel was imaged 20 mm downstream from the inlet alternating between two filter sets (Excitation/Emission: Violet/Blue and Green/Red, respectively) in a fluorescence microscope with the focus set to the channel mid height, Fig. 4a-b. First an image of the dextran dye concentration profile (blue) was acquired after which the filter set was changed and a sequence of 10 cell images (red) was recorded with <5 millisecond exposure time, each image containing up to 10 cells. The filter was then switched back to record a second gradient image to observe potential temporal fluctuations in the gradient. The process of recording gradient and cell images was then repeated 10 to 20 times capturing images of >1000 individual cells at a rate of 1–10 cells per second. The capture rate of the camera was set to 1 Hz to ensure that a cell was only measured once given the flow rate during the measurement.

To read out the acoustic impedance of a cell it was first located in the cell image, Fig. 4c and thereafter the dextran dye intensity was read off the gradient images at the corresponding position, Fig. 4d. The gradient images ($I$) were background-subtracted using images of a channel filled with side-inlet-medium (10% iodixanol and no fluorescent tracer) and then normalized with respect to background-subtracted reference images ($I_{max}$) of a channel completely filled with central inlet medium (36% iodixanol and $3\,\mu g\,ml^{-1}$ fluorescent dextran) such that the ratio ($I/I_{max}$) is a number from 0 to 1. The dextran dye intensity was taken to be linear with the iodixanol concentration (Supplementary Fig. 9) for which the acoustofluidic properties are known (Supplementary Fig. 2) and could be calculated through a 2nd degree polynomial (Fig. 4d). After determining the acoustic impedance of the liquid at the cell's iso-acoustic point we assigned that value to the effective acoustic impedance of the cell (Supplementary Note 1, Eq. 5).

**Code availability.** The custom computer code that support the findings of this study are available from the corresponding author upon request.

**Data availability.** The data that support the findings of this study are available from the corresponding author upon request.

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

## Acknowledgements

The authors are grateful for assistance with experiments by Mads Givskov Senstius (Technical University of Denmark), Louise Gren (Lund University) and Filip Sköld (Lund University), for valuable comments from Peter Muller (Technical University of Denmark) and Pelle Ohlsson (Lund University, Sweden), for chip fabrication by Andreas Lenshof (Lund Univesity, Sweden) and for accessing measurement apparatus from Thomas Laurell (Lund University, Sweden). The study was carried out with financial support from the Swedish Research Council (grant no. 2012-6708), the Royal Physiographic Society, Lund, Sweden and the Birgit and Hellmuth Hertz' Foundation, Lund, Sweden.

## Author contribution

P.A. H.-W.S. and J.V. did the experimental design. P.A. and H.-W.S. performed and analyzed the experiments. P.A., J.T.K. and H.B. made the physical interpretations. J.T.K. and H.B. derived the physical scaling arguments. P.A. and J.T.K. performed numerical modelling. P.A. prepared the manuscript. J.V. and H.B. supervised the project. P.A., J.T.K., H.B. and J.V. edited the manuscript.

## Additional information

**Competing financial interests**: Although not closely related to the herein presented results P.A. declares part-time activity as a consultant in acoustofluidic research and development.

**How to cite this article**: Augustsson, P. *et al.* Iso-acoustic focusing of cells for size-insensitive acousto-mechanical phenotyping. *Nat. Commun.* 7:11556 doi: 10.1038/ncomms11556 (2016).

