## [Peer review file · Nature Communications]

Reviewers' Comments:

Reviewer #1 (Remarks to the Author)

In describing a method to measure the acoustic impedance of single cells, including the possibility to make the measurement on large numbers of cells, the MS provides an interesting description of an ingenious adaptation of otherwise well-known techniques. Based on the use of an unusual cell medium chosen specifically for its acoustic properties, the authors take the reader through many aspects of the method by which this allows the acoustic impedance of cells to be determined and present data which relate to validation of its accuracy.

Whilst this is indeed of interest, the paper has two serious shortcomings.

First, it does not address the issue that, examined at the level of a single cell, acoustic impedance varies spatially within the cell. Whilst it is acceptable to treat a cell as of a uniform (effective) acoustic impedance when considering many cells or homogeneous tissue, it is possible to theorise that the value of the information gained in measuring the acoustic impedance of a single cell will be much reduced if the cell is treated as homogenous when it manifestly is not. A useful reference (from the acoustical imaging community rather than the authors' acoustical manipulation community) is Weiss et al, IEEE Trans. UFFC, 2007.

Second, and not unrelated, the authors present measurements from a range of readily accessible cells as a series of technical examples rather than demonstrating real biological or clinical value in the technique they have implemented. In other words, they have proved that what they propose is possible but not that the measurement of acoustic impedance actually has any value to researchers in the life sciences or medicine. Whilst the measurement of acoustic impedance is a smart trick, especially with the ingenious technique the authors have developed and explained eloquently, the real issues is whether it is of any value and this has not been demonstrated.

That the first criticism is valid is all the more surprising because the authors themselves explicitly link mechanical phenotyping to the internal structure of the cells in the abstract.

With those criticisms registered, the authors have nevertheless very effectively covered many of the bases of their technique, with careful experiments and generally appropriately powered experiments. However, it is frustrating that they do not provide more detail of the acoustical fields which are a key component in the effects they observe. Indeed, they make the extremely basic mistake of defining the acoustical fields in terms of "function generator voltage amplitude" and characterise its effects through observation of the behaviour of beads in a flow.

Generally, a proper definition of acoustical source outputs includes their frequency and pressure in Pa and, whilst the "averaged acoustic energy density" related to the function generator voltage amplitude is not entirely devoid of quantitative value, it is hard to see how a researcher could systematically and quantitatively reproduce the present results without engaging in trial and error experiments in an attempt to mimic those presented. In turn, this lack of quantitation calls into question the degree to which modelling may be considered useful.

There is a similar lack of clarity about acoustical streaming. It is mentioned as a subsidiary effect in a few places, and indeed called into action to explain some results, and there is no suggestion that it is a cause of any specific errors in the present work but, without further detail on the devices the authors use, this is difficult to verify.

A further question must also be raised concerning validation of the data. The results presented in Fig. 5 are undoubtedly of interest but the authors state definitively that certain cells types have higher or lower acoustic impedances than others, albeit with appropriate reference to ensembles of results from many cells. However, when they compare their data with other measurements, specifically of MCF7 cells and RBCs, they do not find exact correspondence, instead noting differences which are of the same order of magnitude as the differences they find between cell types. This suggests they see their new technique as a "gold standard" which may be used as a reference for the future, whereas there appears enough doubt in its measurements that definitive cross-calibration should be performed.

The appearance of slight naivety of their approach is further pointed up by their comparison between cell and macroscopic tissue acoustic impedances, several of the latter of which are entirely irrelevant (e.g. fat, skin) and others which are meaningless owing to lack of definition of what is being measured (e.g. heart, brain). Clear confounding factors such as lack of homogeneity suggest that coincidence of cell and macroscopic impedance are of no value as a comparison.

The field in which the authors are working is a difficult one, and the interface between physical and life sciences requires great care to avoid straying into vague or over-ambitious statements. The present authors are to be commended generally on their precision but the last two paragraphs of

the discussion contain statements which must be removed (thus weakening the impetus to publish) or supported further, specifically the authors' belief that IAF will allow users "to assess cell state and disease progress" and that "the precise spatiotemporal control offered by microfluidics" will lead to new, size-insensitive tools.

To improve this work to the extent that it justifies publication outside field-specific journals (Lab on a Chip etc.) or those specialising in new instruments (Review of Scientific Instruments etc.), the authors should first shift the focus from the new tool they have built to the uses to which it can be put in life sciences, with either at least one demonstration of how it meets a previously unmet need or clear validation that the results it achieves are quantitatively correct through comparison with other measurements. They should also define the device itself more exactly, dealing properly and quantitatively with the question of the acoustical fields it generates and the effects of streaming, heating and other artefactual processes that bedevil many microphoresis techniques.

The references that are provided generally well cover the fields to which the paper relates. However, if the authors are to have it accepted for publication, they will need to include additional references on either comparative quantitative techniques, and their results, or a relevant unmet need in life sciences or medicine. Similarly, the paper is generally highly lucid but stretches the importance of the technique that has been developed a little too far with too little evidence, thus calling up vague and unsubstantiated statements that reduce its credibility.

Reviewer #2 (Remarks to the Author)

Acoustic radiation forces are potentially attractive for cell characterisation due to their dependence on inherent mechanical cell properties, including density and compressibility, making label-free mechanophenotyping possible. Their use in practice has been severely limited by the strong dependence of the radiation forces on the cell size and it is this, rather than mechanical properties, that dominates the attempted characterisation.

This paper overcomes the issue of size by using a medium with an acoustic impedance gradient. The gradient is established by taking advantage of recent observations from Deshmukh et al. regarding the stabilisation of two phase flows within acoustic fields. The current authors also demonstrate that the cells will migrate through the impedance gradient to the point at which the impedance of the cell matches the local impedance of the medium.

The work is novel in both the ability to use acoustophoresis to characterise cells independent of size and in the practical application of an acoustic gradient to stabilise such a laminated flow. The impact

of the technique for mechanophenotyping of cells is potentially significant. The methodology is sound and the results are discussed appropriately, although in the supplementary file key elements of fig 4 are rather small and the graph in fig 5f is largely obscured by the legend.

The paper is clearly written and logically structured and I believe it has the quality, novelty, and importance for publication subject to a few clarifications as follows:

i) The manuscript suggests that the iodixanol concentrations used to create supplementary figs 4 a and b are 20% and 30% ("Repeating this experiment, but adding an extra 10% iodixanol in the center layer" to what was previously 20%), but the supplementary figure caption states "30% and 10%". This needs to be clarified.

ii) In the cell trajectory analysis with single concentration fluid there is clear evidence of substantial acoustic streaming, yet the analysis assumes that establishing of the gradient is diffusion-dominated. Further there is a statement in supplementary fig 4 caption that "... the ultrasound ... counteracts both gravitational relocation and acoustic streaming in the bulk ...". This needs further discussion. Can streaming be ignored because of relative timescales in the flow-through device (in which case provide details) or is there experimental evidence available to show that streaming is suppressed by the presence of a second phase laminated fluid - Deshmukh et al. allude to streaming briefly but don't seem to look at it in detail just note that it isn't observed to broaden the laminated flow. In any case saying that it's the "ultrasound" that "counteracts" acoustic streaming would seem hard to justify.

A few minor points:

Abstract - Should be "...a method for providing a suitable acoustic..."

Ref 29 - check initials for Shields

Ref 38 - "Lord Rayleigh" not Rayleigh L."

Supplementary

p3 - the impedance ratio (Z tilde) isn't defined.

p9 - Two occurrences of "that start out" should be "that starts out"

Reviewer #3 (Remarks to the Author)

"Iso-acoustic focusing of cells for size-insensitive acousto-mechanical phenotyping"

This paper describes the use of gradients in the acoustic impedance of the suspension media to characterise the acoustic impedance of suspended cells. Some of the authors involved have worked on the use of negative/positive acoustic contrast for cell sorting for a number of years, in these systems the fluid impedance is tuned such that it lies between that of the cells being sorted. The use of an impedance gradient, as described for the first time in this work, is a very significant step forward, and opens a range of exciting possibilities in en-masse cell characterisation. In addition, the data shown has been presented in a convincing and thorough manner. As such in my opinion this paper is worthy of inclusion in Nature communications once the following points have been considered:

- 1) More details about the calculation of the cell velocity (Fig 1b) should be included in the paper itself.
- 2) The section title "Tuning the acoustic contrast of cells" is in my opinion correct but counter-intuitive, the impedance of the medium is being tuned, which does tune to cell contrast. But I think the subtitle could lead the reader to expect that the cells are being modified - which they aren't and such a misconception would be unfortunate as it negates the elegance of the method.
- 3) The stability of the gradient is discussed in the context of gravity, and well characterised experimentally. I think there should be an additional discussion in the context of acoustic streaming, which is said to be present as it affects the zero contrast cells.
- 4) It is right that cites are made to the authors' previous work, as it specifically leads to this work. However, when more general statements are made as to the use of acoustics in microfluidics, I thought a more representative view of the field and its recent breakthroughs could have been given especially given the broad readership of Nature Communications.

We appreciate very much the thorough work and constructive feedback that we received from the three reviewers. Encouraged by the points made we have now revised the manuscript such that it better communicates our results.

Reviewer #1 (Remarks to the Author):

In describing a method to measure the acoustic impedance of single cells, including the possibility to make the measurement on large numbers of cells, the MS provides an interesting description of an ingenious adaptation of otherwise well-known techniques. Based on the use of an unusual cell medium chosen specifically for its acoustic properties, the authors take the reader through many aspects of the method by which this allows the acoustic impedance of cells to be determined and present data which relate to validation of its accuracy.

Whilst this is indeed of interest, the paper has two serious shortcomings.

First, it does not address the issue that, examined at the level of a single cell, acoustic impedance varies spatially within the cell. Whilst it is acceptable to treat a cell as of a uniform (effective) acoustic impedance when considering many cells or homogeneous tissue, it is possible to theorise that the value of the information gained in measuring the acoustic impedance of a single cell will be much reduced if the cell is treated as homogenous when it manifestly is not. A useful reference (from the acoustical imaging community rather than the authors' acoustical manipulation community) is Weiss *et al*, IEEE Trans. UFFC, 2007.

The reviewer makes a good point in that as one measures biological information at finer scales, more information is uncovered. We agree that IAF as demonstrated here cannot render detailed information about the internal variation of the acoustic impedance of the cell because the wavelength is much longer than the size of a single cell. With IAF, we are primarily interested in the increased information that occurs in moving from bulk measurements to single-cell measurements, and, as we show in the paper, the effective acoustic impedance of a cell holds valuable information and constitutes the core of the IAF method. The method is cell-type specific and size insensitive, the throughput is high, it can be applied for suspension cells, it can be combined with optical assessment, and it can in the extension be applied for separation.

We agree that Weiss *et al.* is relevant to include and we have therefore added this to the introduction and we have altered the Introduction (paragraph 3) to read:

[Acoustic characterization of tissue has tremendous value in medicine through various forms of medical ultrasound imaging applications such as sound scattering, attenuation and elasticity^{11, 12}. It is therefore reasonable to assume that acoustic properties of individual cells, which include mass density and adiabatic compressibility, are meaningfully related to their biological phenotype. Mapping of sound velocity and attenuation at sub-cellular resolution has been demonstrated for adherent cells using acoustic microscopy^{13, 14, 15, 16} indicating that the acoustic properties are related to the interior content and structure of a cell. While the population average of the acoustic properties of cells in suspension can be deduced from measurements using standard density and sound velocity meters^{13, 14, 15, 16, 17} the acoustic properties of single suspension cells are largely unknown.]

And we have added a sentence to *Results - IAF Principle* that clearly explains the interpretation of the effective acoustic impedance:

[To a good approximation the IAP is the location at which Z_{med} equals the effective acoustic impedance Z_{cell} of the cell (**Supplementary Note 1**). Since the sound wavelength is in IAF much longer than the size of a single cell the effective acoustic impedance can be interpreted

as a measure of the integral of the interior variations in acoustic properties that has been previously mapped using acoustic microscopy^{14, 15, 16}.]

Second, and not unrelated, the authors present measurements from a range of readily accessible cells as a series of technical examples rather than demonstrating real biological or clinical value in the technique they have implemented. In other words, they have proved that what they propose is possible but not that the measurement of acoustic impedance actually has any value to researchers in the life sciences or medicine. Whilst the measurement of acoustic impedance is a smart trick, especially with the ingenious technique the authors have developed and explained eloquently, the real issue is whether it is of any value and this has not been demonstrated.

We agree with the reviewer that in this manuscript we do not demonstrate real biological or clinical value and that this paper rather demonstrates a first, and quite extensive, proof of concept of a new size-insensitive equilibrium characterization method for acoustic impedance which includes the analysis of subtypes of white blood cells.

It is our belief that this method may become useful in flow cytometry either as a means to confine cells of a certain acousto-mechanical phenotype to a specific region of the flow such that label-based or label free identification become feasible because of the greatly reduced number of cells to analyze. A further application relates to isolation of circulating tumor cells from blood and because we find, in applying this method, that the tumor cell line MCF7 cells have much lower acoustic impedance than all major subgroups of white blood cells.

We have now better highlighted in the *Discussion*, paragraph 3, the application area relating to cancer cell isolation:

[Further, the measurements on white blood cells (**Fig. 5a**) and the MCF7 cancer cells (**Fig. 4h**) indicate that these cancer cell line cells can be separated from blood cells with high purity based on their lower IAP. While no clinical samples has been analyzed in this study, this suggests that IAF can potentially be employed to isolate circulating tumor cells from cancer patient blood.]

That the first criticism is valid is all the more surprising because the authors themselves explicitly link mechanical phenotyping to the internal structure of the cells in the abstract.

The reviewer is correct in that, ultimately, all macroscopic properties result from underlying structure; even acoustic microscopy is examining properties created by ensembles of molecules. Similarly with mechanical assessment, some of which can be informative about the interior structure (e.g., cell deformation), others, like density gradient centrifugation or IAF can tell us something about the average interior content of a cell, the amount of water etc. What is critical in a biological assay is whether meaningful differences are discernable at the chosen resolution of the assay, and we show in Fig. 4 and 5 that cells can be distinguished using effective acoustic impedance.

To better describe the measurement being made, we have changed the first sentence of the abstract to:

[Mechanical phenotyping of single cells is an emerging tool for cell classification, enabling assessment of effective parameters relating to cells' interior molecular content and structure.]

With those criticisms registered, the authors have nevertheless very effectively covered many of the bases of their technique, with careful experiments and generally appropriately powered experiments. However, it is frustrating that they do not provide more detail of the acoustical fields which are a key component in the effects they observe. Indeed, they make the extremely basic mistake of defining the acoustical fields in terms of "function generator voltage amplitude" and characterise its effects through observation of the behaviour of beads in a flow.

Generally, a proper definition of acoustical source outputs includes their frequency and pressure in Pa and, whilst the "averaged acoustic energy density" related to the function generator voltage amplitude is not entirely devoid of quantitative value, it is hard to see how a researcher could systematically and quantitatively reproduce the present results without engaging in trial and error experiments in an attempt to mimic those presented. In turn, this lack of quantitation calls into question the degree to which modelling may be considered useful.

The reviewer points out that providing details of the acoustic fields is important for reproducibility, with which we agree entirely. We have therefore added to the manuscript new data (Supplementary Fig. 5) where we map the detailed shape of the acoustic field in a 1 mm segment of the IAF channel. From this we conclude that the local acoustic resonance can be described by a sinusoidal function that has a half wavelength across the channel and with a pressure amplitude of 0.39 MPa, in agreement with the 1D modeling. The additional characterization is presented in the first paragraph of *Methods – Measuring the acoustic field*. In the following paragraphs of *Methods – Measuring the acoustic field* (former Supplementary Methods 3) we describe how we estimate the average acoustic energy density and pressure amplitude along the entire length of the IAF channel. We have revised this section to clarify the relation between the acoustic energy density and pressure amplitude and we now report the voltage amplitude measured over the piezoceramic transducer instead of the function generator voltage amplitude. The full *Methods – Measuring the acoustic field* section now reads:

[To measure the shape of the local acoustic field along a 1 mm segment of the IAF channel, fluorescent polystyrene microbeads (radius 2.23 μm) were investigated using micro particle image velocimetry (PIV). Particles were suspended in 10 % iodixanol (homogeneous) and injected in the IAF channel and the flow was stopped before turning on the acoustic field for a transducer voltage of 3.5 V. Image sequences were acquired at 4 Hz capturing the motion of the microbeads in the acoustic field and commercial PIV software (Dynamic studio, Dantec Dynamics, Denmark) was used to extract velocity information from the images (Supplementary Fig. 5a) using the method of adaptive correlation. The y-components (transverse to the channel) of the velocity vectors from 5 consecutive frames for three repeated acoustic actuation experiments are plotted in Supplementary Fig. 5b. Data was fitted (black dashed line) to the expression $u_{\text{rad}} = u_0 + \partial_t y$ following Eq. (12) in Supplementary Note 3, using u_0 , E_{ac} and k_y as fitting parameters for a spherical polystyrene particle of radius 2.23 μm , density 1050 kg m^{-3} and compressibility $1.65 \cdot 10^{-10} \text{ Pa}^{-1}$. Medium properties for 10% iodixanol (Supplementary Fig. 1) were used to calculate that $\Phi = 0.199$. The fit resulted in $u_0 = -3.557 \cdot 10^{-7} \text{ m s}^{-1}$ (CI_{95} : $-8.014 \cdot 10^{-7}$, $9.004 \cdot 10^{-8}$), $E_{\text{ac}} = 16.27 \text{ J m}^{-3}$ (CI_{95} : 16.16, 16.39), and $k_y = 8315 \text{ m}^{-1}$ (CI_{95} : 8293, 8336). The fitted acoustic wave vector k_y corresponds to a sound wavelength $\lambda = 2\pi / k_y = 756 \mu\text{m}$ which is indicative of a half wavelength resonance in the 375 μm wide IAF channel. The fitted local acoustic energy density at 3.5V actuation corresponds to a local pressure amplitude within the field of view of $p_a = 0.39 \text{ MPa}$.

To estimate the acoustic pressure amplitude and the average acoustic energy density along the whole length of the acoustic resonator channel, as a function of the transducer voltage amplitude, microbeads suspended in 10% iodixanol were injected through a single inlet at constant flow rate. The final position of the microbeads as they arrive at the end of the channel depends on the acoustic energy density, the size and acoustic properties of the particle, and the viscosity, the flow profile and the acoustic properties of the liquid.³⁹ The

trajectories of polystyrene microparticles in acoustic fields are well characterized in earlier work,^{35,36} and by comparing simulation to experiment we can estimate the average acoustic energy density. The beads were imaged 20 mm downstream from the inlet with the microscope focus is set at the mid height of the channel. By superimposing a sequence of such images, we estimate how far into the channel the outermost beads have moved when entering the interrogation region, Supplementary Fig. 6a-e. Then we run the script iteratively and update the acoustic energies for each voltage setting until the simulations for each case match with the observations experiment, Supplementary Fig. 6f. Thereafter we make a fit for $E_{ac} = \frac{1}{4} \kappa_m p_a^2 = kU^2$, Supplementary Fig. 6g, and conclude that $k = 1.22 \text{ J m}^{-3} \text{ V}^2$ and that $E_{ac}^{3.5V} = 15 \text{ J m}^{-3}$ for a transducer voltage amplitude of 3.5 V.]

We have in all our work since (Barnkob [36]) mapped the acoustic field through its actions on suspended microspheres, which is a well-established and widely acknowledged method that relates to early work on acoustophoresis from the group of Donald Feke (Johnson, D. A., & Feke, D. L. (1995). Methodology for fractionating suspended particles using ultrasonic standing waves and divided flow fields. *Separations Technology*, 5(4), 251–258). In previous work (Augustsson [35]) we mapped the local variations in the acoustic field along a section of an acoustophoresis channel and saw that it indeed varied along the length of the device. Further, we showed that the average acoustic energy density can be determined and related to the applied voltage by observing the distribution of suspended particles of known properties at the end of the channel. As described in *Methods – Measuring the acoustic field* (former Supplementary Methods 3), we use this method to characterize the effect of the acoustic field using test particles of known size, density and compressibility, thus enabling other researchers to accurately reproduce our observations or investigate new modalities.

We have added the corresponding acoustic energy densities and pressure amplitudes to the *Methods - Acoustic actuation* where we previously only stated the oscilloscope readings. Specifically, we now write that:

[The resulting transducer amplitude peak to peak was measured with an oscilloscope to be 6.0 V (leading to an acoustic field amplitude in the channel of $E_{ac} = 43 \text{ J m}^{-3}$, $p_a = 0.66 \text{ MPa}$) in the blood cell measurements and 3.5 V (leading to an acoustic field amplitude in the channel of $E_{ac} = 15 \text{ J m}^{-3}$, $p_a = 0.39 \text{ MPa}$) in the cell line experiments. The function generator was set to make repeated linear sweeps in frequency from 1.900 MHz to 2.100 MHz over periods of 1 ms. This leads to stable operation and a more even acoustic field along the length of the device⁴⁹.]

In microscale acoustofluidics, unfortunately, it is currently not possible to deduce, from even the best characterized source output, exactly which acoustic energy density E_{ac} will be present in the microchannel of interest. What is well established is that the acoustically-induced velocity of a suspended particle scales linearly with E_{ac} , which is proportional to the square of the acoustic pressure amplitude p_a and thus to the square of the applied piezoceramic voltage amplitude [Augustsson Ref. 35 and Barnkob Ref. 36]. For any given applied frequency, the voltage amplitude can be recorded either by measuring the voltage drop over the piezoceramic transducer with an oscilloscope, or by simply reading the set value for the function generator output voltage amplitude. Because of frequency-dependent impedance mismatch between the electrical output stage of the function generator and the piezoceramic transducer, these two voltages are most often not identical, but their ratio remains constant. Moreover, the transducer frequency is swept linearly in time from 1.9 MHz to 2.1 MHz (*Methods - Acoustic actuation*), which adds to the complexity of predicting the actual E_{ac} . The relevant figure of merit to report is therefore, as detailed above, the average acoustic energy density E_{ac} actually present in the microchannel. The average acoustic energy density in the channel is in this work between 0 and 44 J/m^3 depending on the configuration of function generator and amplifier and this corresponds to pressure amplitudes between 0 and 0.66 MPa.

There is a similar lack of clarity about acoustical streaming. It is mentioned as a subsidiary effect in a few places, and indeed called into action to explain some results, and there is no suggestion that it is a cause of any specific errors in the present work but, without further detail on the devices the authors use, this is difficult to verify.

This is an interesting topic that has been highlighted by all three reviewers that is worthy of further elaboration. Encouraged by this we have now added a paragraph to *Results – Creating an acoustic impedance gradient*:

[As was pointed out in the previous section, acoustic streaming is observed for cells of zero acoustic contrast in the case of homogeneous medium. In the inhomogeneous system we observed an iodixanol concentration profile that evolved over more than 34 s (Supplementary Fig. 4a) which indicates that acoustic streaming is not present in the bulk to the extent that it affects the acoustic impedance gradient. We explain this absence of streaming in the bulk by a scaling argument in Supplementary Note 2. In essence, the acoustic force density stabilizing the gradient can be shown to be orders of magnitude larger than the shear-force density associated with the boundary-driven acoustic streaming.]

The result that impedance profile can be sustained over more than 34 seconds, and it flattens on the same timescale as in the control experiment for diffusion only (Supplementary Figs. 4a and 4c) indicates that there is no substantial mixing due to acoustic streaming. This is not simply because we cannot detect the streaming, because when it is present, we detect streaming, such as in the case of homogeneous acoustic impedance (Supplementary Movie 2). Acoustic streaming emanates in these systems from the boundary at the top and bottom of the channel, and we have revised and renamed Supplementary Note 2 and added a scaling argument showing that this boundary-layer-driven streaming in the bulk is suppressed in inhomogeneous fluids:

[The experiments show that boundary-driven acoustic streaming in the bulk (Rayleigh streaming) is suppressed in an inhomogeneous fluid. This can be understood by a scaling argument comparing f_{ac} to the magnitude f_{str} of the shear-force density associated with the streaming flow rolls driven by the usual slip-velocity $v_{str} = \Psi \frac{v_a^2}{c_s}$ at the walls. Here, v_a is the amplitude of the acoustic velocity field, c_s is the speed of sound, and Ψ is a geometrical prefactor, which is equal to 3/8 for a planar wall³. A scaling estimate for the shear-force density is then,

$$|f_{str}| \approx \eta \nabla^2 v_{str} \approx \eta \frac{1}{L^2} \Psi \frac{v_a^2}{c_s} \approx \frac{4\Psi\eta}{\rho_{m0}c_sL^2} E_{ac}, \quad (9)$$

where L is the characteristic length scale, and it has been used that $E_{ac} \approx \frac{1}{4} \rho_{m0} v_a^2$. The ratio of the destabilizing streaming force $|f_{str}|$ in Eq. (9) and the stabilizing acoustic force $|f_{ac}|$ in Eq. (8) using $L = h/4$ becomes

$$\frac{|f_{str}|}{|f_{ac}|} \approx \frac{|f_{str}|}{E_{ac}\delta/L} \approx \frac{4\Psi\eta}{\rho_{m0}c_sL\delta} \approx \frac{16\Psi\eta}{\rho_{m0}c_sh\delta} \approx 10^{-4}. \quad (10)$$

Thus the scaling calculation shows that acoustic streaming is suppressed in the bulk of inhomogeneous fluids due to the density-gradient-induced acoustic force f_{ac} .]

A further question must also be raised concerning validation of the data. The results presented in Fig. 5 are undoubtedly of interest but the authors state definitively that certain cells types have higher or lower acoustic impedances than others, albeit with appropriate reference to ensembles of results from

many cells. However, when they compare their data with other measurements, specifically of MCF7 cells and RBCs, they do not find exact correspondence, instead noting differences which are of the same order of magnitude as the differences they find between cell types. This suggests they see their new technique as a "gold standard" which may be used as a reference for the future, whereas there appears enough doubt in its measurements that definitive cross-calibration should be performed.

We believe there is currently no gold standard for measuring the absolute effective acoustic impedance of individual suspension cells, in part because there are no appropriate calibration particles available with which to ascertain the accuracy and precision of different techniques. Our method is an equilibrium method, which provides an interesting alternative to existing dynamic methods. We agree that cross-calibration would be of great value and therefore, in the *Discussion*, 2nd paragraph, we compare with previous findings by other groups that have used other approaches. In our opinion, in the case of the MCF7 cells, there is currently no way of determining which method measures most accurately. As the reviewer points out, however, we find consistent differences between cell types with our approach.

The appearance of slight naivety of their approach is further pointed up by their comparison between cell and macroscopic tissue acoustic impedances, several of the latter of which are entirely irrelevant (e.g. fat, skin) and others which are meaningless owing to lack of definition of what is being measured (e.g. heart, brain). Clear confounding factors such as lack of homogeneity suggest that coincidence of cell and macroscopic impedance are of no value as a comparison.

We find it valuable to point out that the acoustic impedances that we measure on individual cells fall in the same range as available macroscopic measurements for tissue. We think that this observation is interesting to many readers. While not proof of accuracy, individual cell results far outside the range of tissue measurements would raise questions as to the relevance of comparing cells based on their acoustic impedance. This was unclear in the original manuscript, so we have edited paragraph 3 (former paragraph 2) of the *Discussion* to clarify this, which now reads:

[We find it relevant to further investigate cell IAP as a potential biomarker since it was found to be cell-type specific with measured effective acoustic impedances ranging from 1.55 MPa·s/m to 1.75 MPa·s/m (Fig. 4h and Fig. 5a). We find it interesting, to note the close correspondence to literature values for measured acoustic impedances of various tissues (fat 1.38 MPa·s/m, brain 1.60 MPa·s/m, heart 1.45 MPa·s/m, kidney 1.65 MPa·s/m, blood 1.66 MPa·s/m, liver 1.69 MPa·s/m, skin 1.99 MPa·s/m)⁴⁴.].

The field in which the authors are working is a difficult one, and the interface between physical and life sciences requires great care to avoid straying into vague or over-ambitious statements. The present authors are to be commended generally on their precision but the last two paragraphs of the discussion contain statements which must be removed (thus weakening the impetus to publish) or supported further, specifically the authors' belief that IAF will allow users "to assess cell state and disease progress" and that "the precise spatiotemporal control offered by microfluidics" will lead to new, size-insensitive tools.

We see a significant shift in acoustic impedance when comparing untouched leukocytes to cells that has been exposed to RBC lysis buffer and thus we conclude that these cells have undergone a detectable change to a different biophysical state. The second-to-last paragraph of the *Discussion* has been revised such that the statement about detecting the cell state is now supported by pointing out this result, and it has been clearly separated from the speculation about monitoring disease progress. The paragraph now reads:

[Cell deformability has been extensively studied for suspension cells as well as adherent cells and is informative about interior structure³. In contrast, IAF relates to the whole-cell, or effective, compressibility and density, and we believe it renders complementary acousto-mechanical information. As we have seen the effective impedance of neutrophils was dependent on exposure to lysis buffer and we can therefore speculate that IAF can be useful for assessing cell state and disease progress. Microfluidic single cell density measurements have been previously demonstrated to enable sensitive monitoring of yeast growth cycle⁴⁵ and altered cell state resulting from environmental perturbations⁵. We envision that IAF can be informative in a similar way but benefit from higher throughput and flexibility in terms of sample preparation and integration with complementary microfluidic operations. IAF could for instance be combined with label-free or label-based optical analyzers enabling an iso-acoustic tunable band-pass gate within which we interrogate cells of interest.]

Further, we have added references to papers (Squires [46] and Toner [47]) to buttress the statement about microfluidics offering precise spatiotemporal control and we have moderated the last sentence of the *Discussion* such that we now envision a *tunable cell profiling method* rather than a *high precision* one. Throughout the manuscript we state clearly what is supported by data or references on the one hand, and what our vision is on the other hand.

To improve this work to the extent that it justifies publication outside field-specific journals (Lab on a Chip etc.) or those specialising in new instruments (Review of Scientific Instruments etc.), the authors should first shift the focus from the new tool they have built to the uses to which it can be put in life sciences, with either at least one demonstration of how it meets a previously unmet need or clear validation that the results it achieves are quantitatively correct through comparison with other measurements. They should also define the device itself more exactly, dealing properly and quantitatively with the question of the acoustical fields it generates and the effects of streaming, heating and other artefactual processes that bedevil many microphoresis techniques.

The reviewer raises points that we acknowledge and we have therefore revised the manuscript such that it now better communicates how this method can be useful. As discussed previously, we supply new data of for the acoustic field that details the action of the acoustic field on objects within one microscope field of view and we provide new and previously unpublished scaling arguments regarding the acoustic streaming.

The references that are provided generally well cover the fields to which the paper relates. However, if the authors are to have it accepted for publication, they will need to include additional references on either comparative quantitative techniques, and their results, or a relevant unmet need in life sciences or medicine. Similarly, the paper is generally highly lucid but stretches the importance of the technique that has been developed a little too far with too little evidence, thus calling up vague and unsubstantiated statements that reduce its credibility.

We have referenced and highlighted better the connection between mapping of acoustic properties at subcellular resolution and whole-cell measurements. As for the unmet needs, we indicate in the start of the introduction that faster and more precise classification or separation of cells is desirable and that microfluidics offers means to do so. This is supported by two recent publications that address two clinically relevant applications, real-time label-free blood component analysis (Otto [1]) and CTC isolation (Ozkumur [2]), respectively. We think that the revised manuscript better communicates how this method may find use in this context. In particular, as we point out, how our method overcomes the central problem of cell-size-sensitivity in existing acoustophoretic methods.

Reviewer #2 (Remarks to the Author):

Acoustic radiation forces are potentially attractive for cell characterisation due to their dependence on inherent mechanical cell properties, including density and compressibility, making label-free mechanophenotyping possible. Their use in practice has been severely limited by the strong dependence of the radiation forces on the cell size and it is this, rather than mechanical properties, that dominates the attempted characterisation.

This paper overcomes the issue of size by using a medium with an acoustic impedance gradient. The gradient is established by taking advantage of recent observations from Deshmukh et al. regarding the stabilisation of two phase flows within acoustic fields. The current authors also demonstrate that the cells will migrate through the impedance gradient to the point at which the impedance of the cell matches the local impedance of the medium.

The work is novel in both the ability to use acoustophoresis to characterise cells independent of size and in the practical application of an acoustic gradient to stabilise such a laminated flow. The impact of the technique for mechanophenotyping of cells is potentially significant. The methodology is sound and the results are discussed appropriately, although in the supplementary file key elements of fig 4 are rather small and the graph in fig 5f is largely obscured by the legend.

We agree and have updated Supplementary Figs. 4 and 5.

The paper is clearly written and logically structured and I believe it has the quality, novelty, and importance for publication subject to a few clarifications as follows:

i) The manuscript suggests that the iodixanol concentrations used to create supplementary figs 4 a and b are 20% and 30% ("Repeating this experiment, but adding an extra 10% iodixanol in the center layer" to what was previously 20%), but the supplementary figure caption states "30% and 10%". This needs to be clarified.

The figure caption is correct and the main text is now corrected such that it is clear that the central inlet iodixanol concentration is 30% and the side inlets have 10% concentration.

ii) In the cell trajectory analysis with single concentration fluid there is clear evidence of substantial acoustic streaming, yet the analysis assumes that establishing of the gradient is diffusion-dominated. Further there is a statement in supplementary fig 4 caption that "... the ultrasound ... counteracts both gravitational relocation and acoustic streaming in the bulk ...". This needs further discussion. Can streaming be ignored because of relative timescales in the flow-through device (in which case provide details) or is there experimental evidence available to show that streaming is suppressed by the presence of a second phase laminated fluid - Deshmukh et al. allude to streaming briefly but don't seem to look at it in detail just note that it isn't observed to broaden the laminated flow. In any case saying that it's the "ultrasound" that "counteracts" acoustic streaming would seem hard to justify.

This is an interesting topic that has been highlighted by all three reviewers that is worthy of further elaboration. Encouraged by this we have now added a paragraph to *Results – Creating an acoustic impedance gradient*:

[As was pointed out in the previous section, acoustic streaming is observed for cells of zero acoustic contrast in the case of homogeneous medium. In the inhomogeneous system we observed an iodixanol concentration profile that evolved over more than 34 s (**Supplementary Fig. 4a**) which indicates that acoustic streaming is not present in the bulk to the extent that it affects the acoustic impedance gradient. We explain this absence of

streaming in the bulk by a scaling argument in Supplementary Note 2. In essence, the acoustic force density stabilizing the gradient can be shown to be orders of magnitude larger than the shear-force density associated with the boundary-driven acoustic streaming.]

The result that impedance profile can be sustained over more than 34 seconds, and it flattens on the same timescale as in the control experiment for diffusion only (Supplementary Figs. 4a and 4c) indicates that there is no substantial mixing due to acoustic streaming. This is not simply because we cannot detect the streaming, because when it is present, we detect streaming, such as in the case of homogeneous acoustic impedance (Supplementary Movie 2). Acoustic streaming emanates in these systems from the boundary at the top and bottom of the channel, and we have revised and renamed Supplementary Note 2 and added a scaling argument showing that this boundary-layer-driven streaming in the bulk is suppressed in inhomogeneous fluids:

[The experiments show that boundary-driven acoustic streaming in the bulk (Rayleigh streaming) is suppressed in an inhomogeneous fluid. This can be understood by a scaling argument comparing f_{ac} to the magnitude f_{str} of the shear-force density associated with the streaming flow rolls driven by the usual slip-velocity $v_{str} = \Psi \frac{v_a^2}{c_s}$ at the walls. Here, v_a is the amplitude of the acoustic velocity field, c_s is the speed of sound, and Ψ is a geometrical prefactor, which is equal to $3/8$ for a planar wall. A scaling estimate for the shear-force density is then,

$$|f_{str}| \approx \eta \nabla^2 v_{str} \approx \eta \frac{1}{L^2} \Psi \frac{v_a^2}{c_s} \approx \frac{4\Psi\eta}{\rho_{m0}c_s L^2} E_{ac}, \quad (9)$$

where L is the characteristic length scale, and it has been used that $E_{ac} \approx \frac{1}{4} \rho_{m0} v_a^2$. The ratio of the destabilizing streaming force $|f_{str}|$ in Eq. (9) and the stabilizing acoustic force $|f_{ac}|$ in Eq. (8) using $L = h/4$ becomes

$$\frac{|f_{str}|}{|f_{ac}|} \approx \frac{|f_{str}|}{E_{ac} \delta / L} \approx \frac{4\Psi\eta}{\rho_{m0}c_s L \delta} \approx \frac{16\Psi\eta}{\rho_{m0}c_s h \delta} \approx 10^{-4}. \quad (10)$$

Thus the scaling calculation shows that acoustic streaming is suppressed in the bulk of inhomogeneous fluids due to the density-gradient-induced acoustic force f_{ac} .]

A few minor points:

Abstract - Should be "...a method for providing a suitable acoustic..."

Amended accordingly

Ref 29 - check initials for Shields

Ref 38 - "Lord Rayleigh" not Rayleigh L."

Amended accordingly

Supplementary

p3 - the impedance ratio (Z tilde) isn't defined.

Amended accordingly

p9 - Two occurrences of "that start out" should be "that starts out"

Amended accordingly

Reviewer #3 (Remarks to the Author):

"Iso-acoustic focusing of cells for size-insensitive acousto-mechanical phenotyping"

This paper describes the use of gradients in the acoustic impedance of the suspension media to characterise the acoustic impedance of suspended cells. Some of the authors involved have worked on the use of negative/positive acoustic contrast for cell sorting for a number of years, in these systems the fluid impedance is tuned such that it lies between that of the cells being sorted. The use of an impedance gradient, as described for the first time in this work, is a very significant step forward, and opens a range of exciting possibilities in en-masse cell characterisation. In addition, the data shown has been presented in a convincing and thorough manner. As such in my opinion this paper is worthy of inclusion in Nature communications once the following points have been considered:

1) More details about the calculation of the cell velocity (Fig 1b) should be included in the paper itself.

The reviewer correctly points out that we do not show in the main text how the velocity plotted in Fig 1b is calculated. While the plot is indeed based on a calculation of the velocity of a hypothetical cell and using experimental data for the concentration profile, the plot serves here as a conceptual representation of the IAF principle. To clarify this we have revised the Figure 1 caption such that it is clear that this is a conceptual plot.

[(b) Conceptual plot showing that when the acoustic impedance Z_{cell} (dashed blue line) of a given cell matches Z_{med} (full blue line) at the IAP, its transverse velocity u_{rad} (green line) becomes zero so that its position along y reflects its individual effective acoustic impedance.]

Further details about the acoustic radiation force and the velocity and trajectory of a cell are given in Supplementary Notes 1 and 3.

2) The section title "Tuning the acoustic contrast of cells" is in my opinion correct but counter-intuitive, the impedance of the medium is being tuned, which does tune to cell contrast. But I think the subtitle could lead the reader to expect that the cells are being modified - which they aren't and such a misconception would be unfortunate as it negates the elegance of the method.

This is a good point and we have changed the section title to "Tuning the acoustic contrast between medium and cells"

3) The stability of the gradient is discussed in the context of gravity, and well characterised experimentally. I think there should be an additional discussion in the context of acoustic streaming, which is said to be present as it affects the zero contrast cells.

This is an interesting topic that has been highlighted by all three reviewers that is worthy of further elaboration. Encouraged by this we have now added a paragraph to *Results – Creating an acoustic impedance gradient*:

[As was pointed out in the previous section, acoustic streaming is observed for cells of zero acoustic contrast in the case of homogeneous medium. In the inhomogeneous system we observed an iodixanol concentration profile that evolved over more than 34 s (**Supplementary Fig. 4a**) which indicates that acoustic streaming is not present in the bulk to the extent that it affects the acoustic impedance gradient. We explain this absence of

streaming in the bulk by a scaling argument in Supplementary note 2. In essence, the acoustic force density stabilizing the gradient can be shown to be orders of magnitude larger than the shear-force density associated with the boundary-driven acoustic streaming.]

The result that impedance profile can be sustained over more than 34 seconds, and it flattens on the same timescale as in the control experiment for diffusion only (Supplementary Figs. 4a and 4c) indicates that there is no substantial mixing due to acoustic streaming. This is not simply because we cannot detect the streaming, because when it is present, we detect streaming, such as in the case of homogeneous acoustic impedance (Supplementary Movie 2). Acoustic streaming emanates in these systems from the boundary at the top and bottom of the channel, and we have revised and renamed Supplementary Note 2 and added a scaling argument showing that this boundary-layer-driven streaming in the bulk is suppressed in inhomogeneous fluids:

[The experiments show that boundary-driven acoustic streaming in the bulk (Rayleigh streaming) is suppressed in an inhomogeneous fluid. This can be understood by a scaling argument comparing f_{ac} to the magnitude f_{str} of the shear-force density associated with the streaming flow rolls driven by the usual slip-velocity $v_{str} = \Psi \frac{v_a^2}{c_s}$ at the walls. Here, v_a is the amplitude of the acoustic velocity field, c_s is the speed of sound, and Ψ is a geometrical prefactor, which is equal to $3/8$ for a planar wall. A scaling estimate for the shear-force density is then,

$$|f_{str}| \approx \eta \nabla^2 v_{str} \approx \eta \frac{1}{L^2} \Psi \frac{v_a^2}{c_s} \approx \frac{4\Psi\eta}{\rho_{m0}c_sL^2} E_{ac}, \quad (9)$$

where L is the characteristic length scale, and it has been used that $E_{ac} \approx \frac{1}{4} \rho_{m0} v_a^2$. The ratio of the destabilizing streaming force $|f_{str}|$ in Eq. (9) and the stabilizing acoustic force $|f_{ac}|$ in Eq. (8) using $L = h/4$ becomes

$$\frac{|f_{str}|}{|f_{ac}|} \approx \frac{|f_{str}|}{E_{ac}\delta/L} \approx \frac{4\Psi\eta}{\rho_{m0}c_sL\delta} \approx \frac{16\Psi\eta}{\rho_{m0}c_sh\delta} \approx 10^{-4}. \quad (10)$$

Thus the scaling calculation shows that acoustic streaming is suppressed in the bulk of inhomogeneous fluids due to the density-gradient-induced acoustic force f_{ac} .]

4) It is right that cites are made to the authors' previous work, as it specifically leads to this work. However, when more general statements are made as to the use of acoustics in microfluidics, I thought a more representative view of the field and its recent breakthroughs could have been given especially given the broad readership of Nature Communications.

We agree in part. We have added a recent breakthrough of single cell trapping using surface acoustic waves (Collins [24]) and we have added a reference to cell concentration using acoustofluidics (Carugo [22]). The general reference to acoustofluidics is the introductory announcement for a Lab on a Chip tutorial series comprising 23 papers that includes texts from many different groups and authors throughout the acoustofluidic community. We would like to point out that the authors to this paper are from three separate institutions and are working or have previously been working in collaboration with many different groups and are therefore on the author lists on papers from various labs.

Reviewers' Comments:

Reviewer #1 (Remarks to the Author)

The comments which follow relate to those made previously (Reviewer 1).

The authors have carefully addressed all the comments made and the paper is now just about entirely satisfactory, with the caveat that applications for the balancing concept in acoustic tweezing, whilst now acceptable, remain a point of weakness in the overall presentation.

The only, very minor remaining request is that a note is added that special issues apply to the acoustic impedances of fat and skin, which are the only ones out of the range of the authors' measurements, with fat not corresponding with a conventional cellular structure and skin being multi-layered and inaccessible to acoustic (ultrasound) characterisation in terms of acoustic impedance at conventional frequencies.

We again express our gratitude to the time and effort spent by the reviewer and have made a correction to the manuscript based on the points made.

REVIEWERS' COMMENTS:

Reviewer #1 (Remarks to the Author):

The comments which follow relate to those made previously (Reviewer 1).

The authors have carefully addressed all the comments made and the paper is now just about entirely satisfactory, with the caveat that applications for the balancing concept in acoustic tweezing, whilst now acceptable, remain a point of weakness in the overall presentation.

OK

The only, very minor remaining request is that a note is added that special issues apply to the acoustic impedances of fat and skin, which are the only ones out of the range of the authors' measurements, with fat not corresponding with a conventional cellular structure and skin being multilayered and inaccessible to acoustic (ultrasound) characterisation in terms of acoustic impedance at conventional frequencies.

We agree with the reviewer and have made changes to the third paragraph of the discussion so that it now reads

We find it relevant to further investigate cell IAP as a potential biomarker since it was found to be cell-type specific with measured effective acoustic impedances ranging from 1.55 MPa s m⁻¹ to 1.75 MPa s m⁻¹ (Fig. 4h and Fig. 5a). Although organs and tissue typically consist of multiple cell types which can in turn have fundamentally different internal structure than the cells under investigation here we find it interesting to note the close correspondence to literature values for measured acoustic impedances of various tissues (fat 1.38 MPa s m⁻¹, brain 1.60 MPa s m⁻¹, heart 1.45 MPa s m⁻¹, kidney 1.65 MPa s m⁻¹, blood 1.66 MPa s m⁻¹, liver 1.69 MPa s m⁻¹, skin 1.99 MPa s m⁻¹)⁴⁴ with fat, heart and skin being outside our established range. Further, the measurements on white blood cells (Fig. 5a) and the MCF7 cancer cells (Fig. 4h) indicate that these cancer cell line cells can be separated from blood cells with high purity based on their lower IAP. While no clinical samples has been analyzed in this study, this suggests that IAF can potentially be employed to isolate circulating tumor cells from cancer patient blood.